# A host basal transcription factor is a key component for infection of rice by TALE-carrying bacteria

Meng Yuan[1]*, Yinggen Ke[1], Renyan Huang[1], Ling Ma[1], Zeyu Yang[1], Zhaohui Chu[2], Jinghua Xiao[1], Xianghua Li[1], Shiping Wang[1]*

[1]National Key Laboratory of Crop Genetic Improvement, National Center of Plant Gene Research (Wuhan), Huazhong Agricultural University, Wuhan, China; [2]State Key Laboratory of Crop Biology, Shandong Provincial Key Laboratory of Agricultural Microbiology, Shandong Agricultural University, Taian, China

**Abstract** Transcription activator-like effectors (TALEs) are sequence-specific DNA binding proteins found in a range of plant pathogenic bacteria, where they play important roles in host-pathogen interactions. However, it has been unclear how TALEs, after they have been injected into the host cells, activate transcription of host genes required for infection success. Here, we show that the basal transcription factor IIA gamma subunit TFIIAγ5 from rice is a key component for infection by the TALE-carrying bacterium *Xanthomonas oryzae* pv. *oryzae*, the causal agent for bacterial blight. Direct interaction of several TALEs with TFIIAγ5 is required for activation of disease susceptibility genes. Conversely, reduced expression of the *TFIIAγ5* host gene limits the induction of susceptibility genes and thus decreases bacterial blight symptoms. Suppression or mutation of *TFIIAγ5* can also reduce bacterial streak, another devastating disease of rice caused by TALE-carrying *X. oryzae* pv. *oryzicola*. These results have important implications for formulating a widely applicable strategy with which to improve resistance of plants to TALE-carrying pathogens.

*For correspondence: myuan@ mail.hzau.edu.cn (MY); swang@ mail.hzau.edu.cn (SW)

**Competing interests:** The authors declare that no competing interests exist.

## Introduction

Transcription activator-like effectors (TALEs) are important effectors of plant pathogenic bacteria of the genus *Xanthomonas* (*Boch et al., 2009*). The bacteria inject TALEs via their Type III secretion system (T3SS) into host cells, where they translocate to the nucleus and bind host gene promoters in a sequence-specific manner. The DNA binding domain consists of variable repeats that together account for a predictable DNA recognition code (*Boch et al., 2009*; *Moscou and Bogdanove, 2009*). This property has been exploited for programmable DNA binding, and has allowed targeted genome editing by combining TALE DNA binding domains with nucleases (TALENs) (*Maggio and Gonçalves, 2015*). TALE-like proteins are not restricted to the genus *Xanthomonas*, and have also been found in the plant pathogen Ralstonia solanacearum (*de Lange et al., 2014*), and in the endosymbiont *Burkholderia rhizoxinica* (*de Lange et al., 2014*; *Juillerat et al., 2014*). TALE-like proteins thus may play not only antagonistic roles in host-microbe interactions.

Xanthomonas infect many important crops including barley, bean, brassica, cassava, citrus, cotton, mango, pepper, rice, rye, tomato, triticale, and wheat (*Schornack et al., 2013*; *Boch et al., 2014*). In rice, *Xanthomonas oryzae* pv. *oryzae* (*Xoo*) causes bacterial blight and *X. oryzae* pv. *oryzicola* (*Xoc*) causes bacterial streak, both of which are highly devastating diseases. The recessive resistance gene *xa5* is widely used to improve rice resistance to *Xoo* (*Kottapalli et al., 2007*). *xa5* is a natural allele of the gene for the transcription factor IIA gamma subunit 5 (TFIIAγ5), changing a valine to a glutamine (TFIIAγ5^{V39E} thereafter) (*Iyer and McCouch, 2004*; *Sugio et al., 2007*). TFIIA is

**eLife digest** Around the world, bacterial infections reduce the yields of many important crops like rice, tomatoes, peppers and citrus fruits. *Xanthomonas* is a particularly widespread genus of bacteria; it consists of almost 30 species that cause diseases in more than 400 plant hosts, including bacterial blight and bacterial streak in rice plants.

Plants do have an immune system that is able to detect invading microbes and trigger a defensive response against them; however, many disease-causing bacteria have evolved ways to avoid or counteract this response. For example, at least five *Xanthomonas* species use proteins called transcription activator-like effectors (or TALEs for short) to infect their host plants. The bacterial proteins are essentially injected into the plant's cells where they activate specific plant genes that make the host more susceptible to infection. Like other organisms, plants use proteins called transcription factors to switch genes on or off. However, it was not clear if the TALEs hijack the plant's transcriptional machinery to activate these "susceptibility genes" or if they activate the genes via some other means.

Now, Yuan et al. show that TALE-carrying bacteria do make use of at least one of rice's own transcription factors to cause bacterial blight and bacterial streak. The transcription factor in question is rice's version of a general transcription factor, called TFIIAγ, which is essential for gene activation in plants, animals and fungi. Yuan et al. also identify the region of the TALE that binds to the transcription factor, and show that rice plants with lower levels of the transcription factor are protected against bacterial blight and bacterial streak.

Uncovering how disease-causing *Xanthomonas* bacteria use TALEs to infect plants will hopefully help researchers to develop crop plants that are more resistant to these harmful bacteria. Further work is now needed to see if the gene that encodes TFIIAγ in crop plants can be edited to achieve this goal, or whether genes encoding resistant variants of the protein already exist in other plant species.

a basal transcription factor of eukaryotes and it is essential for polymerase II–dependent transcription (*Høiby et al., 2007*). It consists of two subunits, the large subunit TFIIA$\alpha\beta$ and the small subunit TFIIAγ (*Li et al., 1999*).

Rice TFIIAγ5 has been suggested to be a cofactor that directly enables TALEs to induce host gene expression (*Iyer-Pascuzzi and McCouch, 2007*), either as a helper of TALE function (*Boch et al., 2014*), or as a TALE-targeted host gene (*Gu et al., 2009*). The latter scenario is supported by the finding that the TALE PthXo7 directly activates expression of another TFIIAγ encoding gene, *TFIIAγ1* (*Sugio et al., 2007*).

In this paper, we reveal that TALEs from two *Xanthomonas* pathogens, *Xoo* and *Xoc* directly interact with TFIIAγ5 to activate host susceptibility genes, and that RNAi-mediated suppression or mutation of *TFIIAγ5* confers disease resistance. Our results suggest that modifying host *TFIIAγ* genes by mutation or suppression may provide a widely applicable strategy to improve plant resistance to TALE-carrying pathogens.

## Results

### TFIIAγ5 is required for TALE-dependent induction of host genes

To assess whether host TFIIAγ is required for TALE-regulated transcriptional activation of rice susceptibility genes, we first assessed how pair of rice near-isogenic lines, IR24 carrying *TFIIAγ5* and IRBB5 carrying mutant *TFIIAγ5$^{V39E}$* in the IR24 background, responded to 15 different TALE-carrying *Xoo* strains (*Yang and White, 2004*). IRBB5 always showed fewer disease symptoms than IR24 (*Figure 1—figure supplement 1A*). *Xoo* infection did not induce RNA expression of *TFIIAγ5* in IR24 or *TFIIAγ5$^{V39E}$* in IRBB5 (*Figure 1A*), which correlates with the absence of predicted DNA binding motifs for known TALEs in the *TFIIAγ5* promoter. In contrast, expression of known disease susceptibility genes Os*8N3*, *TFIIAγ1*, Os*TFX1*, and Os*11N3*, each of which is targeted by a different TALE (*Römer et al., 2010*; *Sugio et al., 2007*; *Yang et al., 2006*), was always lower in IRBB5 (p<0.01),

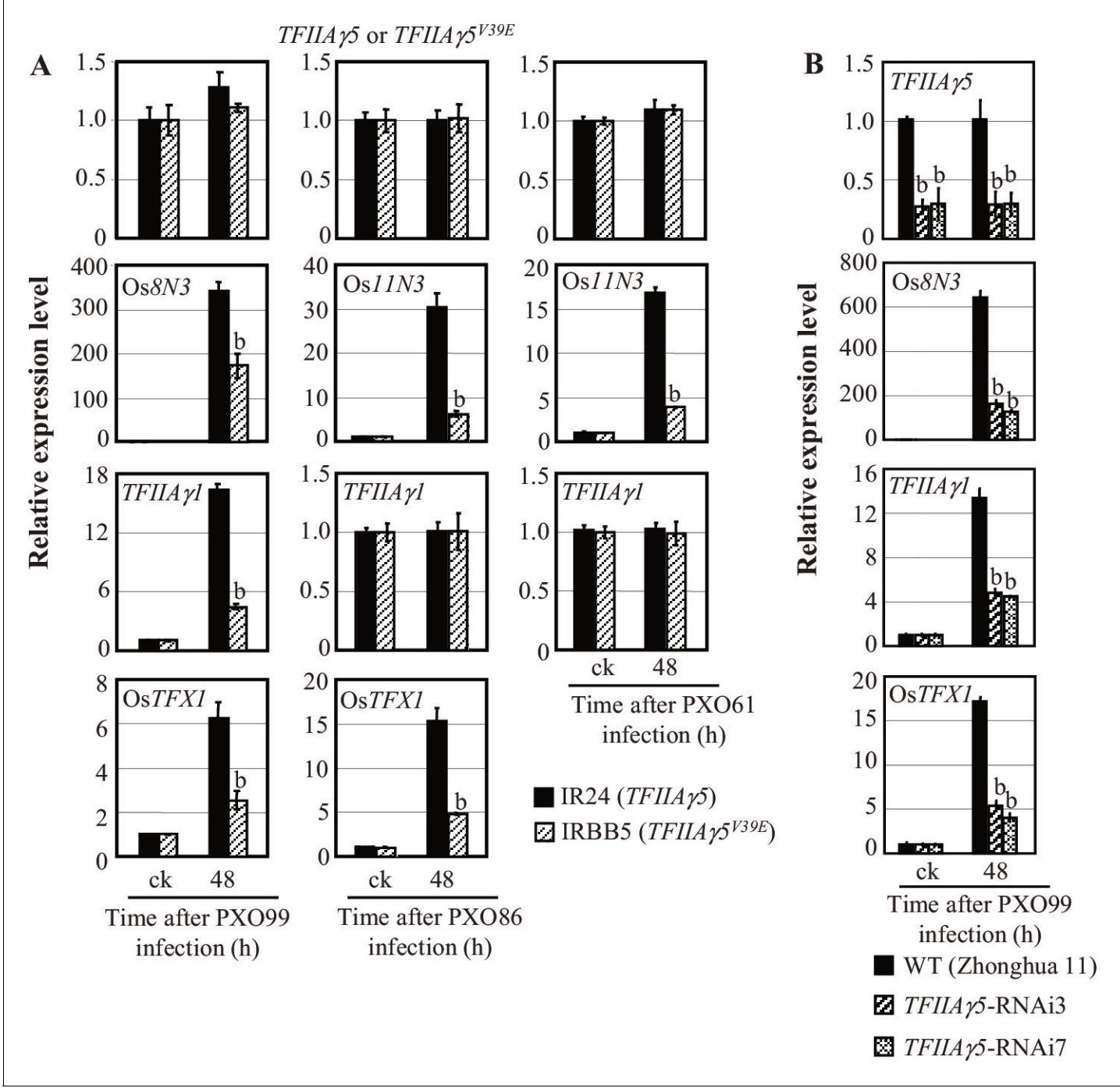

**Figure 1.** Effects of *TFIIAγ5* on the expression of disease susceptibility genes Os*8N3*, Os*11N3*, *TFIIAγ1*, or Os*TFX1,* after *Xoo* infection. Plants were inoculated with *Xoo* strain PXO99 (harbouring TALEs PthXo1, PthXo7, and PthXo6), PXO86 (harbouring TALE PthXo3) or PXO61 (harbouring TALE AvrXa7) at the booting (panicle development) stage. It is known that PthXo1, PthXo7, and PthXo6 induce Os*8N3*, *TFIIAγ1*, and Os*TFX1*, respectively, and PthXo3 and AvrXa7 all induce Os*11N3*. Each bar represents mean (three replicates) ± standard deviation. (**A**) Mutation of *TFIIAγ5* (rice line IRBB5). b, significant difference between IR24 and IRBB5 at p<0.01. (**B**) *TFIIAγ5*-RNAi lines. b, significant difference between wild-type (WT) and transgenic plants at p<0.01.

The following figure supplements are available for figure 1:

**Figure supplement 1.** Effects of *TFIIAγ5* on rice resistance to *Xoo* strains known to carry TALEs.

**Figure supplement 2.** Effect of *TFIIAγ5* on *Xa23*-mediated resistance to *Xoo* strain PXO99.

although not necessarily completely abolished (*Figure 1A*). Together, these results point to TFIIAγ5 being a host co-factor for TALE-dependent induction of susceptibility genes.

To determine directly the role of *TFIIAγ5* in host gene expression, we suppressed its activity by RNA interference (RNAi). Only the expression of *TFIIAγ5*, but not of *TFIIAγ1* was reduced in $T_0$ transgenic plants, and the reduction in *TFIIAγ5* expression correlated with enhanced resistance to *Xoo* PXO99 in $T_0$ and $T_1$ plants (*Figure 1—figure supplement 1B,C*). *TFIIAγ5*-RNAi plants also had enhanced resistance to a diverse collection of 13 additional *Xoo* strains (*Figure 1—figure supplement 1D*), and *Xoo*-induced expression of Os8N3 and *OsTFX1* was reduced in *TFIIAγ5*-RNAi plants (*Figure 1B*). Suppressing *TFIIAγ5* did not obviously influence growth and development of the transgenic plants.

TALE DNA-binding motifs have been detected in the promoters of some disease resistance genes, an apparent evolutionary response against TALE-carrying bacteria (*Gu et al., 2005*; *Römer et al., 2010*; *Wang et al., 2015*). *Xoo* TALE AvrXa23 activates the *Xa23* resistance gene, resulting in resistance to *Xoo* (*Wang et al., 2015*). To investigate the role of *TFIIAγ5* in *Xa23* resistance, we crossed rice lines IRBB5, with a *xa23* susceptibility and a *TFIIAγ5^{V39E}* resistance allele, and CBB23, with a *Xa23* resistance and a *TFIIAγ5* susceptibility allele. $F_2$ plants of genotypes *Xa23/Xa23* or *Xa23/xa23* were completely resistant to PXO99 in the *TFIIAγ5/TFIIAγ5* or *TFIIAγ5/TFIIAγ5^{V39E}* background, but showed the reduced resistance in the *TFIIAγ5^{V39E}/TFIIAγ5^{V39E}* background (*Figure 1—figure supplement 2A*). Consistent with the resistance phenotype, *Xa23* expression was rapidly induced by PXO99 in *Xa23/Xa23* or *Xa23/xa23* plants when they also were of genotype *TFIIAγ5/TFIIAγ5* or *TFIIAγ5/TFIIAγ5^{V39E}* (*Figure 1—figure supplement 2B*). *Xa23* induction was completely lost in *TFIIAγ5^{V39E}/TFIIAγ5^{V39E}* plants. These results suggest that TFIIAγ5 plays dual roles in *Xoo*−rice interactions: it is required by TALE-containing *Xoo* to cause disease, but at the same time it can help to protect against disease in the presence of certain resistance genes that have TALE-binding motifs in their promoters.

## *Xoo* TALEs directly interact with TFIIAγ5

*Xoo* TALEs typically have an amino-terminal translocation signal (TS), a central repeat region (RR), a transcription factor binding (TFB) region, a nuclear localization signal (NLS), and a carboxyl-terminal transcription activation domain (AD) (*Figure 2—figure supplement 1*, *Figure 2—source data 1*) (*Yang et al., 2006*; *Schreiber et al., 2015*). When fused to the GAL4 DNA binding domain, *Xoo* TALE PthXo1 on its own could activate reporter gene expression in yeast. This was observed whenever the TS or AD were present, but not with the RR, TFB or NLS (*Figure 2—figure supplement 1A*). This is similar to what has been reported for *Xoo* TALE AvrXa10 and *X. euvesicatoria* TALE AvrBs3 (*Szurek et al., 2001*; *Zhu et al., 1998*).

We hypothesized that TALEs use TFIIAγ5 directly as a co-factor to induce transcription of susceptibility genes. In yeast two-hybrid (Y2H) assays, truncated PthXo1, RR-TFB-NLS, lacking transcriptional activation ability, interacted strongly with TFIIAγ5, somewhat less so with the mutant TFIIAγ5^{V39E}, and not at all with the large subunit of TFIIA, TFIIAαβ (*Figure 2—figure supplement 1B,C*). The interaction with TFIIAγ5 required the TFB (*Figure 2—figure supplement 1D*).

To determine whether this observation of interaction of a TALE TFB with TFIIAγ5, was general, we isolated the TFB encoding DNA fragments from 14 of the 18 other TALE genes in *Xoo* pv. PXO99 (*Salzberg et al., 2008*). These TFBs are 134 to 145 amino acids long, with the Tal7b and Tal8b TFBs predicted to be identical (*Figure 2—source data 2*). All 14 TFB fragments interacted with TFIIAγ5, but only two (Tal7a and Tal8a) with TFIIAγ5^{V39E} (*Figure 2—figure supplement 1E,F*). Notably, different from PthXo1, Tal7a and Tal8a interacted equally well with TFIIAγ5 and TFIIAγ5^{V39E}. The TFBs of Tal7a, Tal8a, and PthXo1 differed by 1 to 20 residues from the other 12 TFBs that interacted only with TFIIAγ5 (*Figure 2—source data 2*).

We confirmed the interactions observed in the Y2H system by transient expression of Myc- and FLAG-labeled proteins in *Nicotiana benthamiana*, followed by co-immunoprecipitation (CoIP) (*Figure 2A*). We found the interaction of full-length PthXo1 with TFIIAγ5 or TFIIAγ5^{V39E}, of the TFBs of PthXo1, PthXo6, PthXo7, Tal3a, Tal7a, and Tal9e with TFIIAγ5, and of the TFBs of PthXo1 and Tal7a with TFIIAγ5^{V39E} (*Figure 2B*).

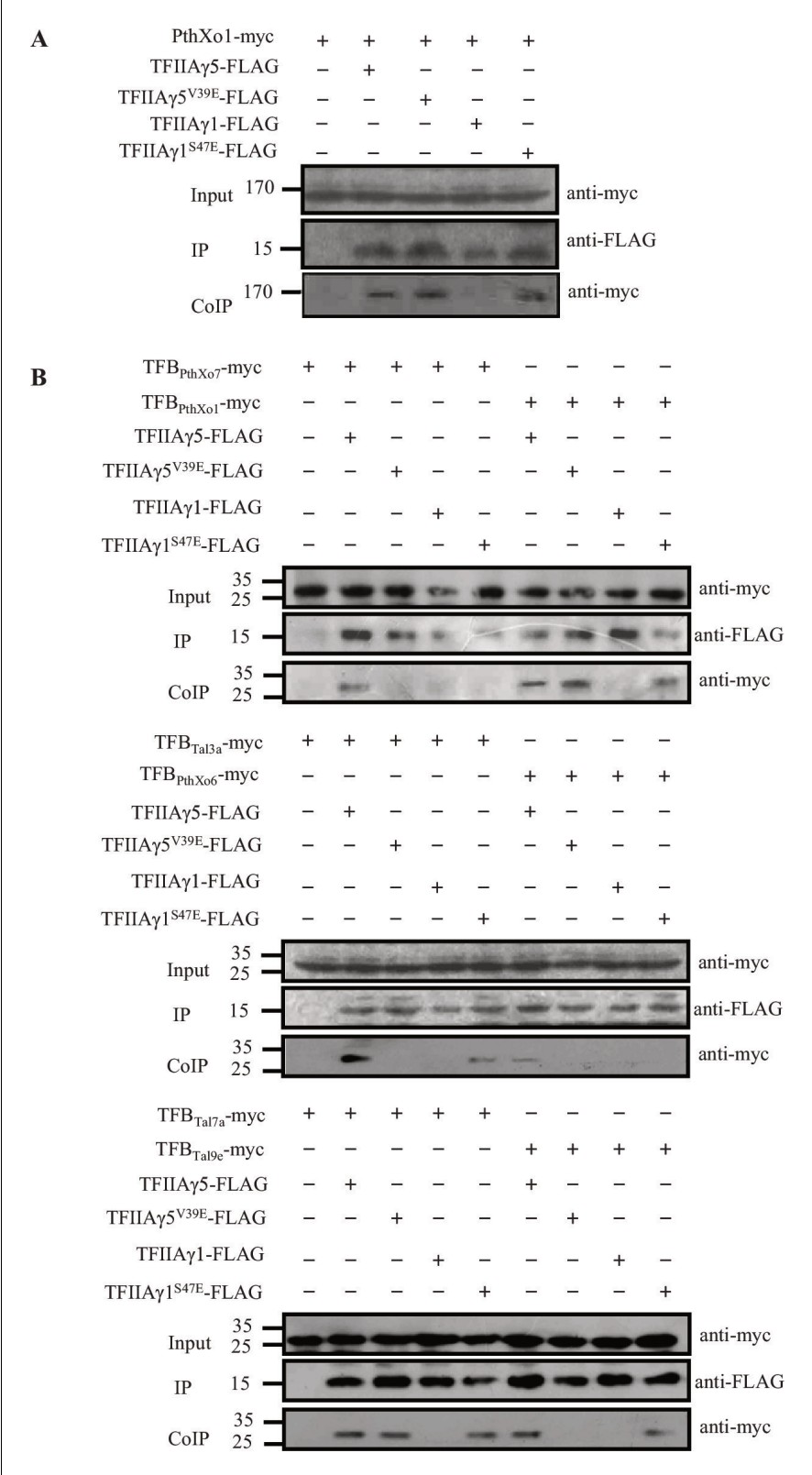

**Figure 2.** Detection of interactions between rice TFIIAγs and TALEs from *Xoo in planta* by co-immunoprecipitation. The protein–protein interaction assays were performed in *N. benthamiana* leaf cells. Proteins before (input) and after immunoprecipitation (IP) were detected with anti-myc and anti-FLAG antibodies. (**A**) Interaction of the myc-labelled full-length PthXo1 with FLAG-labelled TFIIAγ5, TFIIAγ5$^{V39E}$, and mutated rice TFIIAγ1 (TFIIAγ1$^{S47E}$). (**B**) Interactions of the myc-labelled TFB regions of six TALEs with FLAG-labelled rice TFIIAγs.

*Figure 2 continued on next page*

*Figure 2 continued*

The following source data and figure supplement are available for figure 2:

**Source data 1.** The defined domains/motifs and sequences of TALE PthXo1 from *Xoo* strain PXO99.
**Source data 2.** Amino acid sequence alignment of the TFB regions of TALEs from *Xanthomonas oryzae* strains, composed of either 134 or 145 amino acids.
**Figure supplement 1.** Interactions between *Xoo* TALEs and plant TFIIAγs in yeast cells.

## TALE-dependent induction of host genes requires interaction with TFIIAγ5 but not TFIIAγ1

To learn whether the TFB region of TALEs is directly responsible for TALE-induced host gene expression, we generated recombinant *Xoo* strains in which the TFB of PthXo1 was replaced with different TFBs, chosen based on their differential interaction with TFIIAγ5 and TFIIAγ5$^{V39E}$: PthXo1 (TFIIAγ5 > TFIIAγ5$^{V39E}$), Tal7a (TFIIAγ5 = TFIIAγ5$^{V39E}$), and PthXo7 and AvrXa23 (TFIIAγ5 but not TFIIAγ5$^{V39E}$) (*Figure 2—figure supplement 1B,E,F*). In addition, we generated a TFB deletion in PthXo1. The constructs were introduced into *Xoo* pv. T7174 and KACC10331, both of which lack PthXo1. Rice strain IR24, which carries *TFIIAγ5*, is strongly susceptible to T7174 and moderately susceptible to KACC10331, while IRBB5, which carries *TFIIAγ5$^{V39E}$*, is resistant to both *Xoo* strains (*Figure 3A*, *Figure 1—figure supplement 1A*, and *Figure 3—figure supplement 1A*).

As expected, the deletion control PthXo1-*ΔTFB* did not change the success of infection by T7174 or KACC10331 (*Figure 3A*, and *Figure 3—figure supplement 1A*), while the TFBs from PthXo1 and Tal7a, which can interact with both TFIIAγ5 and TFIIAγ5$^{V39E}$, enhanced infection success in both hosts, IR24 (*TFIIAγ5*) and IRBB5 (*TFIIAγ5$^{V39E}$*). Consistent with Tal7a, but not PthXo1, interacting equally well with TFIIAγ5 and TFIIAγ5$^{V39E}$, only the Tal7a TFB caused similar sized lesions in both IR24 and IRBB5 (*Figure 3A*, and *Figure 3—figure supplement 1A*). The TFBs of PthXo7 and AvrXa23, which can interact only with TFIIAγ5, accordingly increased disease symptoms only on IR24. Lesion size in these experiments was correlated with titer of bacterial growth (*Figure 3B*, and *Figure 3—figure supplement 1B*) and expression of Os*8N3* (*Figure 3C*, and *Figure 3—figure supplement 1C*).

The TFB region of the TALEs harbours an imperfect leucine zipper motif, a known protein-protein interaction domain (*Schreiber et al., 2015*). We generated three TFB mutants of PthXo1, substituting leucine with alanine residues (*Figure 3—source data 1A*). The muations did, however, not compromise interaction with TFIIAγ5, nor infection success (*Figure 3—source data 1B,C*).

The other TFIIAγ encoded in the rice genome, TFIIAγ1, shares 86% sequence identity with TFIIAγ5 (*Figure 4—source data 1*), but has a very restricted expression profile, with highest expression in endosperm and stamens (*Figure 4—figure supplement 1*). TFIIAγ1 did not interact with full-length or truncated PthXo1 or other *Xoo* TALE TFBs in yeast or *in planta* (*Figure 2*, and *Figure 2—figure supplement 1B,E,F*).

We produced eight TFIIAγ1 derivatives with TFIIAγ5 substitutions at six positions (*Figure 4—figure supplement 2*). Of 15 TFBs tested, those of PthXo1, Tal3a, Tal7a, Tal8a, Tal9d and Tal9e could interact in yeast with TFIIAγ1$^{S47E}$, but not with other TFIIAγ1 mutants (*Figure 4—figure supplement 2*, and *Figure 2—figure supplement 1E*). Four of these interactions could be confirmed *in planta* (*Figure 2B*).

We then generated *TFIIAγ1*-RNAi plants as well as transgenic plants expressing the *TFIIAγ1$^{S47E}$* mutant from *TFIIAγ1* regulatory sequence. Both types of plants were morphologically normal. Some T$_0$ *TFIIAγ1*-RNAi plants showed enhanced resistance to *Xoo* pv. PXO99 (*Figure 4—figure supplement 3*). Increased resistance was associated with reduced *TFIIAγ1* expression, whereas *TFIIAγ5* expression was unaffected (*Figure 4—figure supplement 3*), which was confirmed in two T$_1$ families (*Figure 4A*). However, these plants did not show enhanced resistance to other 13 *Xoo* strains (*Figure 4B*). This is in agreement with previous suggestions that the *TFIIAγ1* promoter is a target of

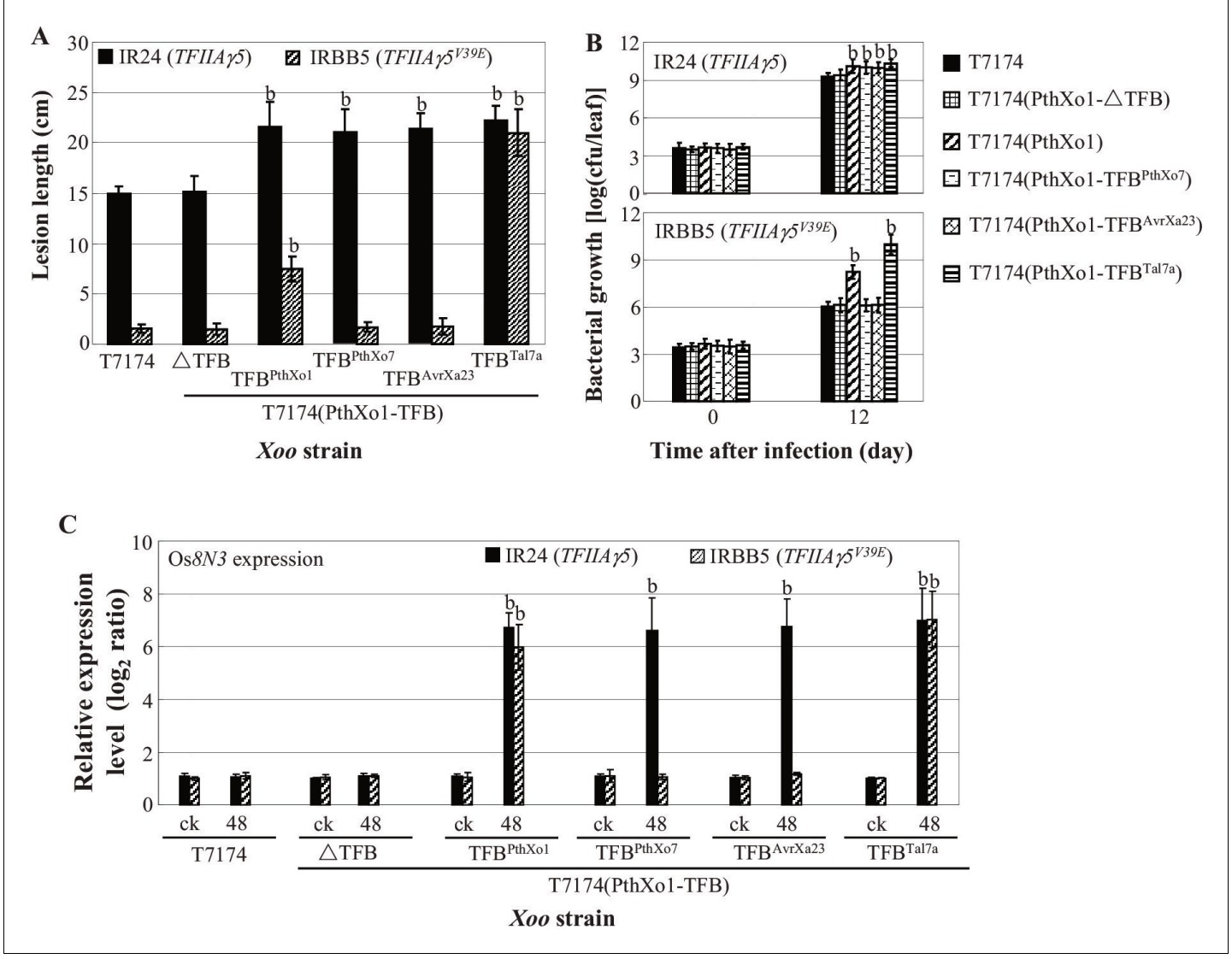

**Figure 3.** Effects of the TFB region of TALE PthXo1 on the virulence of *Xoo* strains and on the expression of rice susceptibility gene in rice—*Xoo* interaction. Each bar represents mean (total 30 to 35 leaves from five plants for lesion length; three replicates for gene expression and bacterial growth rate) ± standard deviation. (**A**) Virulence of wild-type strain T7174 and recombinant strains carrying PthXo1 and its derivatives in IR24 and IRBB5. b, significant difference between T7174 and recombinant strains in each rice line at p<0.01. (**B**) Growth of different *Xoo* strains in rice leaves. b, significant difference between 0 day (30 min after infection) and 12 days after infection of each strain at p<0.01. (**C**) Expression of susceptibility gene Os*8N3* after infection of different strains. b, significant difference between non-inoculated (ck) and inoculated (at 48 hr after infection of a strain) plants in each rice line at p<0.01.

The following source data and figure supplement are available for figure 3:

**Source data 1.** Effects of leucine residues of PthXo1 TFB region on TALE-mediated infection.

**Figure supplement 1.** Effects of the TFB region of TALE PthXo1 on the virulence of *Xoo* strains and on the expression of rice susceptibility gene in rice—*Xoo* interaction.

the TALE PthXo7 from PXO99 (*Boch et al., 2009*; *Sugio et al., 2007*). PthXo7-induced *TFIIAγ1* expression is dependent on *TFIIAγ5* (*Figure 1*).

In the background of *TFIIAγ5^{V39E}*, the *TFIIAγ1^{S47E}*-transgenic plants showed increased susceptibility to *Xoo* pv. PXO99 and PXO341 (*Figure 4C*). The increased susceptibility to PXO99 might be due to an interaction between TFIIAγ1^{S47E} and PthXo1 (*Figure 2B*) to induce the susceptibility gene

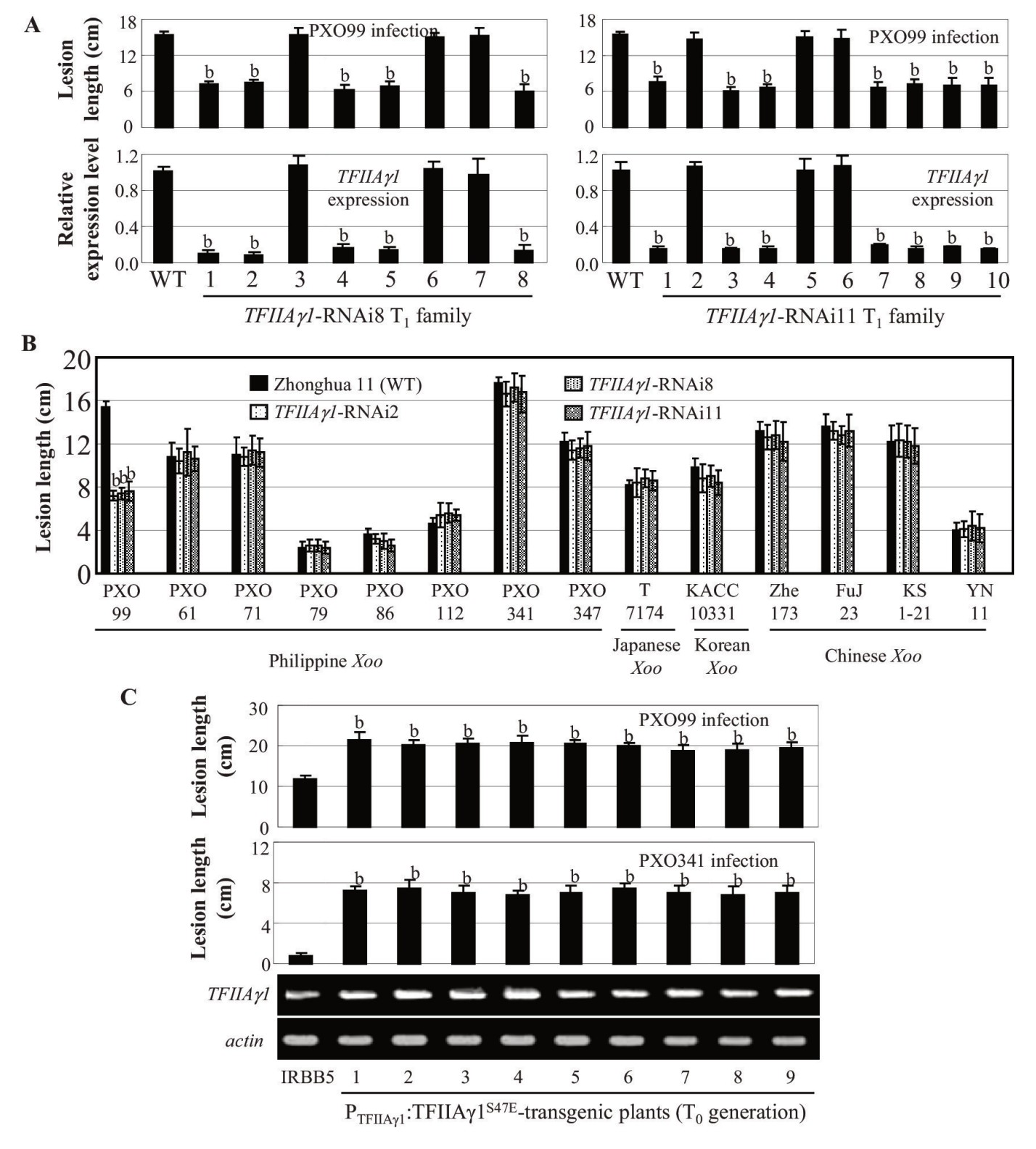

**Figure 4.** Effects of *TFIIAγ1* on response to infections by different *Xoo* strains. Plants were inoculated with *Xoo* at the booting stage. Each bar represents mean (three replicates for gene expression and total 35 to 40 leaves from five plants for lesion length) ± standard deviation. (**A**) Suppressing *TFIIAγ1* enhanced rice resistance to strain PXO99. b, significant difference between wild-type (WT) Zhonghua 11 and transgenic plants at p<0.01. (**B**) Suppressing *TFIIAγ1* did not change rice response to other strains. b, significant difference between WT and transgenic plants at p<0.01. (**C**) P~TFIIAγ1~:
*Figure 4 continued on next page*

*Figure 4 continued*

TFIIAγ1[S47E]-transgenic plants showed susceptibility to PXO99 and PXO341 compared to IRBB5. b, significant difference between IRBB5 and transgenic plants at p<0.01.

The following source data and figure supplements are available for figure 4:

**Source data 1.** Amino acid sequence alignment of basal transcription factor IIA gamma subunit (TFIIAγ) from different species.

**Source data 2.** Single nucleotide polymorphisms in the *TFIIAγ1* coding region of 1419 rice accessions from RiceVarMap (http://ricevarmap.ncpgr.cn).

**Source data 3.** Single nucleotide polymorphisms in the *TFIIAγ5* coding region of 1419 rice accessions from RiceVarMap (http://ricevarmap.ncpgr.cn).

**Figure supplement 1.** Expression profiles of *TFIIAγ5* and *TFIIAγ1* in 28 tissues covering the entire life cycle of rice varieties Minghui 63 and Zhenshan 97.

**Figure supplement 2.** Interactions between the TFB region of TALE PthXo1 and mutated TFIIAγ1s in yeast cells.

**Figure supplement 3.** Effect of suppressing *TFIIAγ1* on rice resistance to *Xoo* strain PXO99.

**Figure supplement 4.** Effect of mutation of *TFIIAγ1* on the expression of disease susceptibility gene Os*8N3* after *Xoo* infection.

Os*8N3* (*Figure 4—figure supplement 4*), while the susceptibility to PXO341 may be explained by another TALE (see the TFBs tested in *Figure 2—figure supplement 1E,F*) that can interact with TFIIAγ1[S47E].

## Genetic variation in *TFIIAγ5* and *TFIIAγ1* genes

We searched the RiceVarMap database of 1419 rice accessions (http://ricevarmap.ncpgr.cn; *Zhao et al., 2015*) for allelic variation at *TFIIAγ1* and *TFIIAγ5*. There were no non-synonymous single nucleotide polymorphisms (SNPs) in *TFIIAγ1* (*Figure 4—source data 2*). Thirty-three rice accessions shared the same two non-synonymous SNPs diagnostic for the *TFIIAγ5[V39E]* allele (*Figure 4—source data 3*). Twenty-nine of these belong to the Aus group, which is mainly from South Asia, and the other four accessions belong to the Indica II group, mainly from Southeast Asia (*Xie et al., 2015*) (*Figure 4—source data 3*). The regional distribution of the *TFIIAγ5[V39E]* resistance allele likely reflects the high disease pressure in these regions.

## *Xoc* TALEs hijack TFIIAγ5 to cause bacterial streak

To learn whether TALEs of other pathogenic bacteria also exploit TFIIAγ5 to cause disease, we investigated the interaction of TFIIAγ5 with TALEs from *Xoc*, which causes bacterial streak. *Xoc* pv. RH3 has at least 11 TALE genes based on DNA blot analysis (*Figure 5—figure supplement 1*). All TFBs of RH3 TALEs (GenBank accession numbers KU163014 to KU163031) interacted with TFIIAγ5 in yeast, and two were confirmed *in planta* (*Figure 5A*, and *Figure 5—figure supplement 2A*). Seven randomly chosen TFBs did not interact with TFIIAγ5[V39E] or TFIIAγ1, but three interacted with TFIIAγ1[S47E] in yeast, and at least one *in planta* (*Figure 5A*, and *Figure 5—figure supplement 2B*). Consistent with these results, rice accession IRBB5 (*TFIIAγ5[V39E]*) was more resistant to infection by different *Xoc* strains than IR24 (*TFIIAγ5*) (*Figure 5—figure supplement 2C*). *TFIIAγ5*-RNAi plants also showed enhanced resistance to *Xoc*, whereas suppressing *TFIIAγ1* had no effect on resistance to *Xoc* (*Figure 5B*).

A recent study has shown that a major quantitative trait locus for resistance to *Xoc* col-localizes with *TFIIAγ5* (*Xie et al., 2014*). Two additional studies have revealed that a TALE that occurs in at least 10 sequenced *Xoc* strains transcriptionally activates the gene for the sulphate transporter Os*SULTR3;6*, a major susceptibility gene in rice−*Xoc* interactions (*Cernadas et al., 2014*; *Wilkins et al., 2015*). *Xoc*-induced expression of Os*SULTR3;6* was significantly reduced (p<0.01) in IRBB5 relative to IR24 (*Figure 5C*), suggesting that TALE-containing *Xoc* also requires TFIIAγ5 to infect rice via TALE-induced expression of host susceptibility genes.

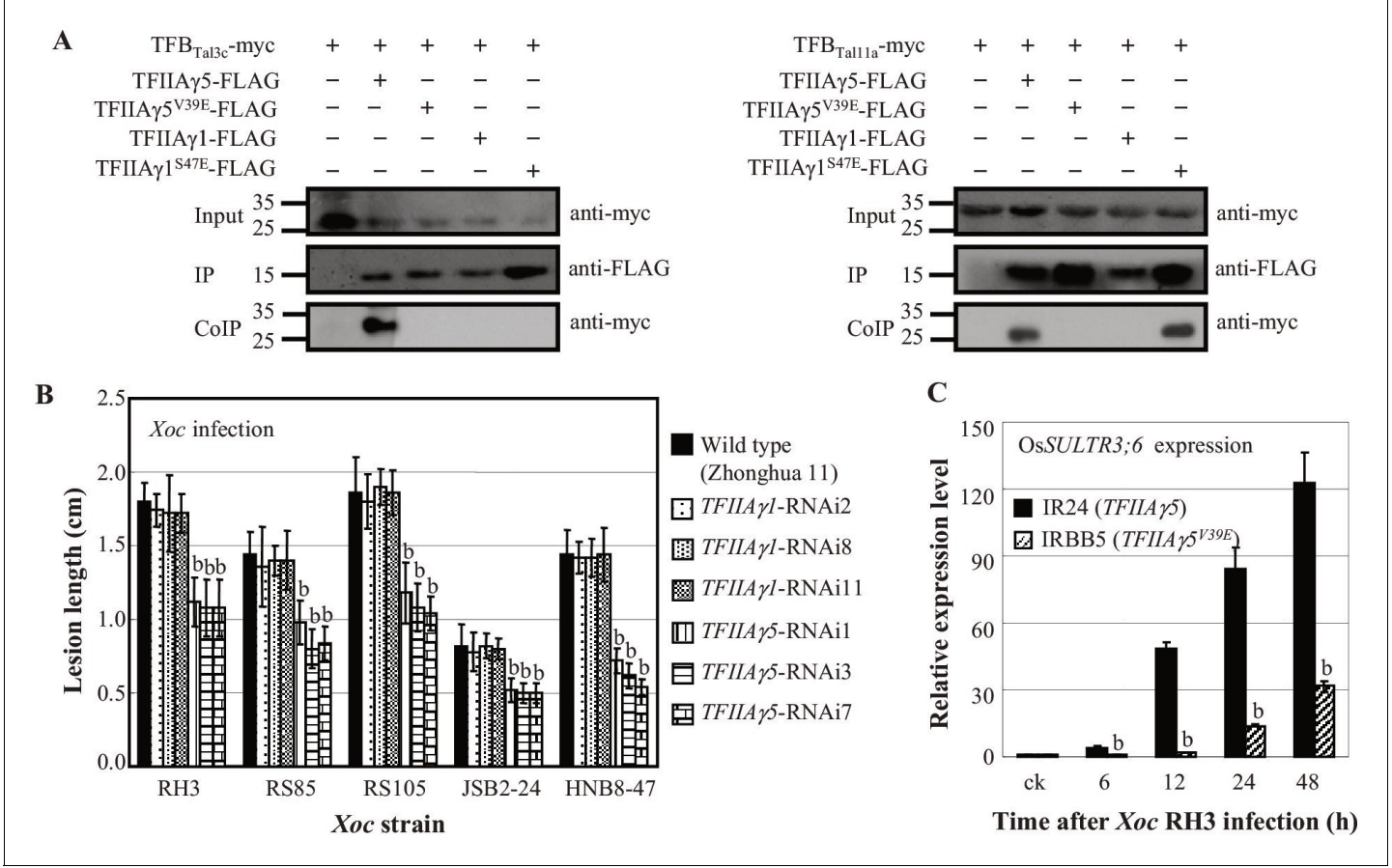

**Figure 5.** Effect of TFIIAγ on rice-*Xoc* interaction. (**A**) Interactions of myc-labelled TFB regions of TALEs from *Xoc* RH3 and FLAG-labelled rice TFIIAγs in *N. benthamiana* leaf cells analysed by CoIP assays. Proteins before (input) and after immunoprecipitation (IP) were detected with anti-myc and anti-FLAG antibodies. (**B**) *TFIIAγ5*-RNAi but not *TFIIAγ1*-RNAi plants showed enhanced resistance to *Xoc* strains. Each bar represents mean (total 30 to 35 leaves from five plants) ± standard deviation. b, significant difference between wild-type and transgenic plants after infection of a strain at p<0.01. (**C**) Mutation of *TFIIAγ5* (rice line IRBB5) reduced expression of disease susceptibility gene Os*SULTR3;6* after infection. Each bar represents mean (three replicates) ± standard deviation. b, significant difference between IR24 and IRBB5 at p<0.01.

The following figure supplements are available for figure 5:

**Figure supplement 1.** Southern hybridization analysis of TALE genes in different *Xanthomonas* species.

**Figure supplement 2.** Analysis of interactions between *Xoc* TALEs and rice TFIIAγs.

## Discussion

TFIIAγ is indispensable for polymerase II–dependent transcription (*Li et al., 1999*). We have shown here how TALE-carrying *Xoo* and *Xoc* exploit rice TFIIAγ5 for activating transcription of downstream susceptibility genes (*Figure 6*). TALEs from *Xoo* and *Xoc* bind to TFIIAγ5 through their TFB regions, and the binding and binding strength are associated with the induction of susceptibility genes. Thus, TFIIAγ5 functions as a key component for TALE-induced host gene expression.

It is striking that the only *TFIIAγ5* paralog in rice, *TFIIAγ1*, apparently functions as a downstream susceptibility gene for *Xoo* PXO99, such that the TALE PthXo7 directly activates *TFIIAγ1* transcription (*Sugio et al., 2007*), which differs from the protein-protein interaction of several *Xoo* TALEs with TFIIAγ5.

The recessive disease resistance allele *TFIIAγ5^{V39E}* confers markedly reduced TALE-dependent induction of downstream susceptibility genes, apparently without compromising the overall activity of TFIIA. The rice accession IRBB5 carrying *TFIIAγ5^{V39E}* is indistinguishable from the near-isogenic

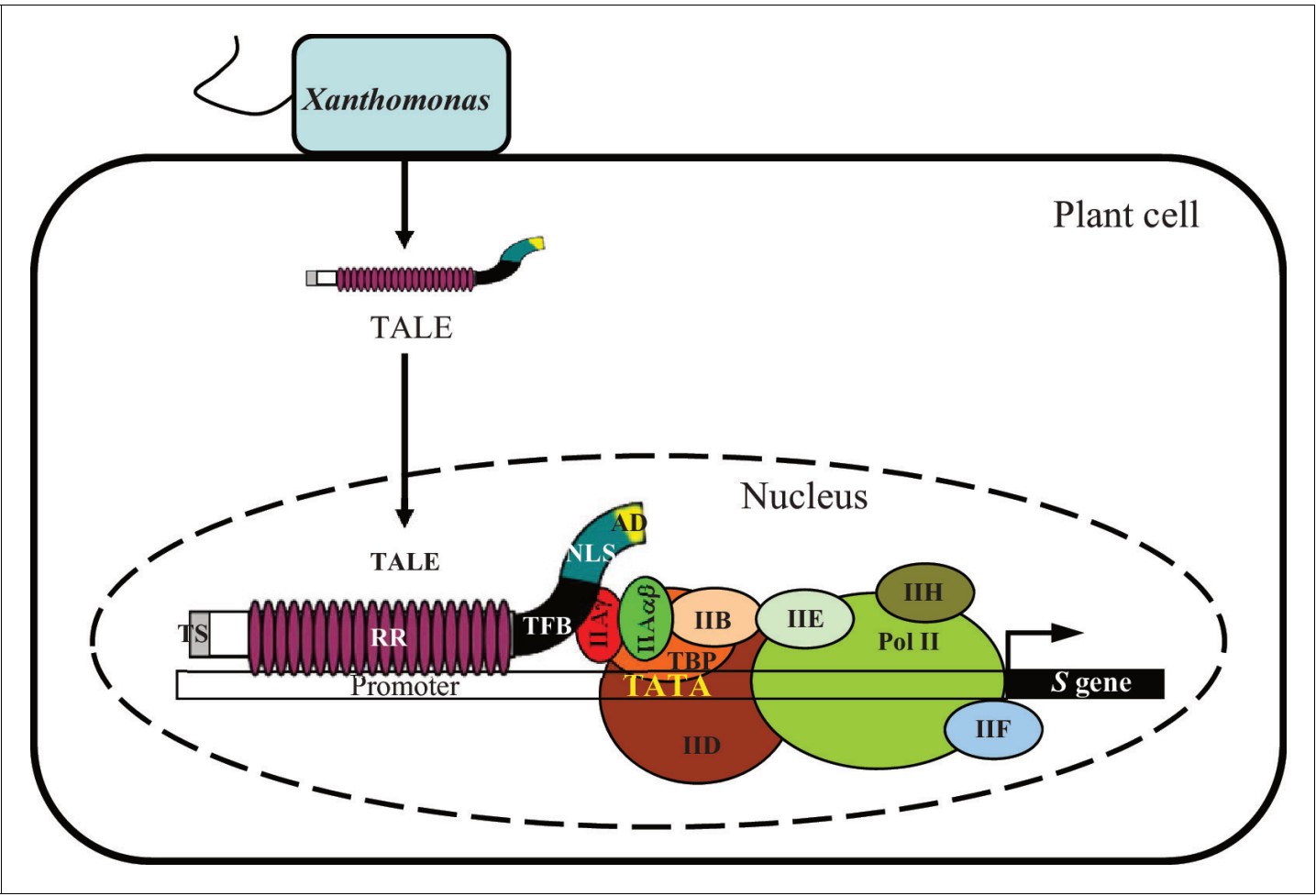

**Figure 6.** A model showing TFIIAγ5 as a key component of rice infection by *Xanthomonas* bacteria. The bacteria hijack rice basal transcription factor TFIIAγ5 (IIAγ) by the transcription factor binding (TFB) region of their TALEs to induce host susceptibility (S) genes for infection. TS, amino-terminal translocation signal; RR, central repeat region; NLS, nuclear localization signal; AD, carboxyl-terminal transcription activation domain. The IIAγ belongs to the transcription pre-initiation complex. This complex consists of transcription factors IIA, which is composed of IIAβα subunit and IIAγ subunit, IIB, IID, IIE, IIF, and IIH, RNA polymerase II (Pol II), and TATA-binding protein (TBP). The binding of transcription pre-initiation complex to the TATA box of promoter was adopted and modified based on *Yudkovsky et al. (2000)*.

line IR24 in plant morphology and agronomic performance, including heading date, flag leaf length, number of panicles per plant, panicle length, grains per panicle, 1000-grain weight, yield per plant, seed setting rate, grain length, width, and thickness, with only slightly reduced plant height (*Supplementary file 1*). Here, we have shown that not only the specific point mutant *TFIIAγ5^{V39E}* has increased *Xoo* resistance, but also that this can also be achieved by RNAi mediated knockdown of *TFIIAγ5*. In addition, we have shown that TALEs from other *Xanthomonas* pathogens, such as *Xoc*, exploit *TFIIAγ5*. Alteration of *TFIIAγ5* activity, either through the introduction of the *TFIIAγ5^{V39E}* allele, or through other reduction-of-function mutations, can provide a general strategy for improving rice resistance to TALE-carrying pathogens. TALE-carrying bacteria cause diseases in many other crops (*Schornack et al., 2013*; *Boch et al., 2014*). If these bacteria also exploit the host TFIIAγ for infection, modification of TFIIAγ may provide a road to improving disease resistance in other crops as well.

# Materials and methods

## Plant and bacterial materials

A pair of near-isogenic lines, IR24 (*TFIIAγ5*) and IRBB5 (*TFIIAγ5^{V39E}*), and the variety Zhonghua 11 were used in this study. Plants were grown during the normal rice growing season under natural field conditions in the Experimental Stations of Huazhong Agricultural University, Wuhan, China.

Chinese, Japanese, Korean, and Pilipino *Xoo* strains were used to study rice resistance to bacterial blight disease (*Gao et al. 2010*; *Li et al., 2012*). Resistance to *Xoc* was tested using Chinese strains (*Ke et al., 2014*). *X. campestris pv. campestris* strain was used for Southern blot analysis of TALE genes (*He et al., 2007*). All *Xanthomonas* strains were grown at 28°C on nutrient agar medium. Antibiotics were used at the following final concentrations as required: ampicillin at 100 μg ml$^{-1}$, rifampicin at 75 μg ml$^{-1}$, kanamycin at 25 μg ml$^{-1}$, and spectinomycin at 50 μg ml$^{-1}$ when genetic manipulation of bacteria.

## Transformation

To construct RNA interference vector, the 3′ untranslated regions of *TFIIAγ5* and *TFIIAγ1* were amplified with primers listed in *Supplementary file 2* and inserted into vector pDS1301 (*Yuan et al., 2010*). *Agrobacterium*-mediated transformation of rice was performed (*Lin and Zhang, 2005*; *Ge et al., 2006*).

## Pathogen inoculation

*Xoo* inoculation described in more detail at Bio-protocol (*Ke et al., 2017*). To evaluate reaction of rice plants to *Xoo*, plants were inoculated with the *Xoo* strains by the leaf-clipping method at the booting (panicle development) stage (*Chen et al., 2002*). The disease was scored by measuring the lesion length at 14 days after inoculation. Each bacterial inoculation assay was repeated at least twice. The disease of some plants was also evaluated by analysing bacterial growth based on a count of the colony-forming units as described previously (*Sun et al., 2004*). For measuring bacterial growth, one *Xoo*-infected leaf from each plant was examined as one replicate, and a total of three plants for each sample were analysed.

To evaluate *Xoc* resistance, rice plants were inoculated with *Xoc* strains by the penetration method using a needleless syringe at the booting stage (*Ke et al., 2014*). Disease was scored by lesion length at 14 days after inoculation. Each bacterial inoculation assay was repeated twice.

## Gene expression analysis

The 2-cm leaf segments next to the bacterial infection sites in the rice plants were collected for RNA isolation. Quantitative reverse transcription-PCR (qRT-PCR) was conducted using gene-specific primers (*Supplementary file 3*) as described previously (*Qiu et al., 2007*). The expression level of the rice *actin* gene was used to normalize the measurement of the expression. Each rice sample was a mixture of leaf tissue from at least five plants, with 8 to 10 leaves per plant. For transgenic plants, segments from three to five leaves of the plant were sampled. Each qRT-PCR assay was repeated at least twice, with each repetition having three technical replicates.

## Vector construction

The TALE PthXo1 was cloned into pHM1 vector to produce pHM1pthXo1, and transferred into *Xoo* strains T7174 and KACC10331 following published method (*Yang and White, 2004*). The TFB region of PthXo1 was replaced with TFB regions of other TALEs by Gibson assembly (*Gibson et al., 2009*). The recombinant strains were confirmed by PCR amplification of TALE fragments.

## Southern hybridization analysis

A standard procedure for Southern hybridization of the bacterial DNA was performed (*Gu et al., 2009*). Genomic DNA from different *Xanthomonas* strains was digested with *Sph*I, separated by electrophoresis on 1.2% agarose gel in TAE buffer, blotted onto a nylon membrane, and hybridized using a $^{32}$P-labeled 2.9-kb *Sph*I fragment of PthXo1.

## Transactivation activity assay

The transactivation activity of PthXo1 was analysed in yeast cells as described previously (*Deng et al., 2012*). The open reading frame of *pthXo1* was ligated into pGBKT7 vector and fused in frame with the yeast GAL4 DNA binding domain. The recombinant vector was transformed into yeast strain AH109. The transformed yeast cells were plated on SD/−Trp or SD/−Trp-His medium and cultured for 3 days as described previously (*Yuan et al., 2010*).

## Protein–protein interaction assay

The interaction between bacterial TALE proteins and host proteins in yeast cells was assayed using MATCHMAKER GAL4 Two-Hybrid System 3 (Clontech, Mountain View, CA) according the manufacturer's instructions. To construct the interaction vectors, full-length and truncated TALEs and the TFB regions of TALEs and plant genes were amplified using the PCR primers listed in *Supplementary file 2*. The amplified DNA fragments were first inserted into vector pBluescript (Agilent Technologies, Santa Clara, CA) for sequencing confirmation. The confirmed bacterial DNA fragments were then ligated into pGBKT7 vector, and the confirmed plant DNA fragments were then ligated into pGADT7 Rec vector. The recombinant pGBKT7 and pGADT7 plasmids were co-transformed into yeast strain AH109 for yeast two-hybrid assays following the lithium acetate method (*Yuan et al., 2010*). The yeast clones were first scribed on the synthetic defined premixes (SD) medium lacking leucine (L) and tryptophan (W) (−LW). The growth of yeast cells on SD/−LW medium indicated that they carried both pGBKT7 and pGADT7 plasmids. An aliquot (10 µl) of 1:10 diluted stationary phase cultured yeast clones grown on the SD/−LW medium was then scribed on the selective SD medium lacking L, W, histidine (H), and adenine (A) (−LWHA). The growth of yeast cells on SD/−LWHA medium indicated that the examined proteins interacted with each other. The interactions of these proteins were also assessed by examination of $\beta$-D-galactopyranoside (X-$\alpha$-gal) activity and $\beta$-galactosidase (LacZ) activity as described previously (*Yuan et al., 2010*). Each yeast two-hybrid assay was repeated at least twice.

CoIP assays were performed to study the interaction between TALE proteins and plant proteins *in planta*. The 9×myc DNA fragment was amplified from pN-TAPa vector (*Rubio et al., 2005*) by using myc-specific primers (*Supplementary file 2*) and inserted into the *Sma*I- and *Bam*HI-digested pU1301 vector (*Cao et al., 2007*), resulting in a vector that we named pU1301-9myc. The DNA fragments of full-length, truncated, or TFB region of TALEs were ligated into the pU1031-9myc vector. The DNA fragments of plant genes were ligated into the pU1301-3FLAG vector (*Yuan et al., 2010*). The recombinant vectors were introduced into *Agrobacterium tumefaciens* strain GV3101. *Agrobacterium*-mediated transformation was performed by infiltration into *N. benthamiana* leaves using a needleless syringe (*Yuan et al., 2010*). CoIP assays were conducted using anti-FLAG antibody (RRID: AB_259529, Sigma-Aldrich, St. Louis, MO) and anti-myc antibody (Tiangen, Beijing, China) as described previously (*Yuan et al., 2010*). Each CoIP assay was repeated at least twice.

## Site-directed mutation

Mutations of plant genes and the *Xoo* TALE genes were made using the GeneTailor Site-Directed Mutagenesis System (Invitrogen Life Technologies, Carlsbad, CA) as described previously (*Yuan et al., 2011*). The mutagenic primers are listed in *Supplementary file 2*.

## Statistical analysis

Differences between samples were analysed for statistical significance by *t*-test in Microsoft Excel (Microsoft, Redmond, WA). Correlations between gene transcript level and disease level were calculated using CORREL analysis in the Microsoft Office Excel program.

## Acknowledgements

We thank Professor Gongyou Chen of Shanghai Jiao Tong University for kindly providing *Xanthomonas* bacterial strains and Professor Qifa Zhang of Huazhong Agricultural University and the eLife editorial team for their help in editing the manuscript. This work was supported by grants from the National Natural Science Foundation of China (31330062, 31100875, and 31371926), the National Program on the Development of Basic Research in China (2012CB114005), the Special Key Project

by the Ministry of Science and Technology of China (2016YFD0100903), and the Fundamental Research Funds for the Central Universities (2014PY039).

## Additional information

### Funding

| Funder | Grant reference number | Author |
|---|---|---|
| National Natural Science Foundation of China | 31100875 | Meng Yuan |
| National Natural Science Foundation of China | 31371926 | Meng Yuan |
| Ministry of Science and Technology of the People's Republic of China | 2012CB114005 | Meng Yuan Shiping Wang |
| Ministry of Science and Technology of the People's Republic of China | 2014PY039 | Meng Yuan |
| Ministry of Science and Technology of the People's Republic of China | 2016YFD0100903 | Shiping Wang |
| National Natural Science Foundation of China | 31330062 | Shiping Wang |

The funders had no role in study design, data collection and interpretation, or the decision to submit the work for publication.

### Author contributions

MY, Designed and performed most of the experiments, Analysed the data, Drafted the manuscript; YK, Helped to generate transgenic rice plants and Xoo mutant, analyse protein-protein interactions and amplify the transcription activator-like effector; RH, LM, ZY, ZC, Helped to generate transgenic rice plants and Xoo mutant, analyse protein-protein interactions, and amplify the transcription activator-like effector; JX, XL, Provided biochemical and molecular analysis support and management and final approval of the manuscript; SW, Supervised the project, Designed some of the experiments, Interpreted data, Revised the manuscript

### Author ORCIDs

Zhaohui Chu, http://orcid.org/0000-0001-8320-7872
Shiping Wang, http://orcid.org/0000-0002-8743-3129

## Additional files

### Supplementary files

• Supplementary file 1. Measurements of agronomic traits of rice lines IR24 and IRBB5 under natural field conditions.

• Supplementary file 2. PCR primers used for construction of vectors for protein–protein interactions and transformation, and detection of positive transgenic plants.

• Supplementary file 3. PCR primers used for quantitative RT-PCR assays.

### Major datasets

The following previously published dataset was used:

| Author(s) | Year | Dataset title | Dataset URL | Database, license, and accessibility information |
|---|---|---|---|---|
| Wang L, Xie W, Chen Y, Tang W, Yang J, Ye R, Liu L, Xu C, Lin Y, Xiao J, Zhang Q | 2010 | Dissecting the developmental transcriptomes of rice | http://www.ncbi.nlm.nih.gov/geo/query/acc.cgi?acc=GSE19024 | Publicly available at the NCBI Gene Expression Omnibus (accession no. GSE19024) |

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
