## [Decision Letter]

Thank you for submitting your work entitled "Modifying host basal transcription factor increases resistance to TALE-carrying bacteria in different crops" for consideration by *eLife*. Your article has been reviewed by three peer reviewers, including Laurent Noel, and the evaluation has been overseen by Thorsten Nürnberger as the Reviewing Editor and Detlef Weigel as the Senior Editor.

The reviewers have discussed the reviews with one another and the Reviewing Editor has drafted this decision to help you prepare a revised submission.

All referees agree that the manuscript has improved and reports compelling evidence that TALE effectors harbor a domain that interacts with TFIIAγ (*Xa5*) thereby activating target gene expression. Furthermore, TALE variants that cannot interact with a given TFIIAγ variant (*xa5*) are affected in their capability to activate a host gene. Altogether, these findings are considered to be a significant advancement in the field that will be of interest to a broad readership.

The manuscript must, however, be substantially revised before final acceptance (including professional language editing). As it stands, it does not easily deliver its message to its readers.

In particular, you need to significantly shorten the manuscript in order to improve its comprehensibility. Further, you should focus solely on data obtained in the rice/*X. oryzae* pathosystem. These data are sufficient on their own, particularly because complementary data obtained from tomato and pepper systems lack comparable scientific rigor and quality. Please remove all figures and text that relates to the pepper and tomato experiments.

It is further mandatory that you qualify the statement that the domain you identified resembles an imperfect leucine zipper (LZ). A zipper consists of two rows of protruding, regularly spaced teeth, which may be made to interdigitate. A leucine zipper refers to a zipper where periodically arranged leucine residues (every 7th aa) constitute the teeth. However, the sequence shown in Figure 1—figure supplement 1 reveals that the so-called leucine zipper is far from having regularly spaced leucine residues: L-X6-L-Χ2-L-X4-L-Χ2-L-Χ2-L-X12-L-X5-L-X25-L-X26-L-Χ2-L-X8-L-X6-L-X10-L. In fact, the "LZ" domain shown counts 14 leucines in a stretch of 134 aa and is thus far from being something similar to an LZ.

[Editors' note: the manuscript was rejected after a second round of peer review, but the authors submitted for reconsideration.]

Thank you for submitting your work entitled "Modifying host basal transcription factor increases resistance to TALE-carrying bacteria in rice" for consideration by *eLife*. Your article has been reviewed by two peer reviewers, and the evaluation has been overseen by Thorsten Nürnberger as the Reviewing Editor and Detlef Weigel as the Senior Editor. Our decision has been reached after consultation between the reviewers. Based on these discussions and the individual reviews below, we regret to inform you that your work will not be considered further for publication in *eLife*.

All referees acknowledge that your study reports on findings that are potentially most interesting to *eLife*'s readership. They agree that content and quality of the science reported have been improved. They unfortunately also agreed that – despite several revisions and suggestions to improve your manuscript – the manuscript is still not accessible to non-expert readers in the field and that the overall quality of presentation falls short of our publication standards.

*Reviewer #1:*

The revised version of the manuscript has been shortened according to the reviewers’ comments. Yet, otherwise the manuscript has not changed significantly in its contents or structure although the reviewers pointed out that *substantial rewriting* would be necessary in order to make this manuscript accessible to a non-TALE specialist.

I am still convinced that the experimental data provided in this manuscript would have deserved publication in a high impact journal like *eLife*. Yet, I am afraid the authors have missed their chance to present their data properly.

I am convinced that professional language editing on its own will not help to improve the manuscript. Rather the authors should team up with other scientist in order to improve the structure of the manuscript.

In sum the current manuscript version is only accessible to TALE insiders and thus seems not appropriate for publication in *eLife*.

*Reviewer #2:*

The manuscript "Modifying host basal transcription factor increases resistance to TALE-carrying bacteria in rice" by Yuan et al. is a revised version of paper previously submitted to *eLife*. The manuscript reports on the recruitment of the general transcription factor TFIIAγ5 to mediate TALE-dependent susceptibility or resistance in rice.

The manuscript has now been significantly streamlined. Most of the comments made in the previous review have been incorporated and the paper is now easier to read (though still very complex).

Here are few re-iterated (or additional) remarks:

It was suggested to simplify qRT-PCR data by only presenting one (or two) of the time points (24 or 48h). This would make data so much easier to read without affecting the message. I do not see the need for a time course and the authors' choice is not clearly justified in their reply.

Abstract is still "cryptic" and should be improved to better catch reader's attention.

Figure 5—figure supplement 1 was modified by deleting two lanes in the Southern blot: Was the Southern redone? If both lanes were merged on the computer, a dashed line should indicate this image manipulation. Or can the authors present raw data for both Southerns? This hold true for any other figure panel.

Westerns for Y2H experiments are not shown. This would be important, as the authors tend to use Y2H to measure interaction strengths. Should be done for some key experiments at least.

*Reviewer #2 (Additional data files and statistical comments):*

Statistical analysis of Q-PCR or pathoassays results by t-test is likely incorrect since it only applies to relatively large data sets (ca. >30 biological replicates) with normal distribution. Were both assumptions verified?

---

## [Author Response]

*All referees agree that the manuscript has improved and reports compelling evidence that TALE effectors harbor a domain that interacts with TFIIAγ (Xa5) thereby activating target gene expression. Furthermore, TALE variants that cannot interact with a given TIIA γ variant (xa5) are affected in their capability to activate a host gene. Altogether, these findings are considered to be a significant advancement in the field that will be of interest to a broad readership.*

*The manuscript must, however, be substantially revised before final acceptance (including professional language editing). As it stands, it does not easily deliver its message to its readers.*

We greatly appreciate your helpful suggestions. The manuscript has been substantially revised based on the suggestions and comments of editors and reviewers. It was also revised by a professional language editing agency.

*In particular, you need to significantly shorten the manuscript in order to improve its comprehensibility. Further, you should focus solely on data obtained in the rice/X. oryzae pathosystem. These data are sufficient on their own, particularly because complementary data obtained from tomato and pepper systems lack comparable scientific rigor and quality. Please remove all figures and text that relates to the pepper and tomato experiments.*

We removed one section entitled “Other TALE carrying bacteria also use host TFIIAγ to cause diseases in different plant” from the Results, which include data obtained from tomato and pepper systems and data obtained by expression of *Arabidopsis* and citrus TFIIAγs in rice. We also removed corresponding text in the Discussion and in the Materials and methods.

It is further mandatory that you qualify the statement that the domain you identified resembles an imperfect leucine zipper (LZ). A zipper consists of two rows of protruding, regularly spaced teeth, which may be made to interdigitate. A leucine zipper refers to a zipper where periodically arranged leucine residues (every 7th aa) constitute the teeth. However, the sequence shown in Figure 1—figure supplement 1 reveals that the so-called leucine zipper is far from having regularly spaced leucine residues: L-X6-L-Χ2-L-X4-L-Χ2-L-Χ2-L-X12-L-X5-L-X25-L-X26-L-Χ2-L-X8-L-X6-L-X10-L. In fact, the "LZ" domain shown counts 14 leucines in a stretch of 134 aa and is thus far from being something similar to an LZ.

I agree with you that this region does have a typical structure of a leucine zipper motif. It does not seem to have the function of a leucine zipper either for none of the leucine residues of this region influences the function of TALE. We changed the ‘imperfect leucine zipper motif’ to “transcription factor binding (TFB) region” in the revised manuscript.

[Editors' note: the author responses to the re-review follow.]

*All referees acknowledge that your study reports on findings that are potentially most interesting to eLife's readership. They agree that content and quality of the science reported have been improved. They unfortunately also agreed that – despite several revisions and suggestions to improve your manuscript – the manuscript is still not accessible to non-expert readers in the field and that the overall quality of presentation falls short of our publication standards.*

*Reviewer #1:*

*The revised version of the manuscript has been shortened according to the reviewers’ comments. Yet, otherwise the manuscript has not changed significantly in its contents or structure although the reviewers pointed out that substantial rewriting would be necessary in order to make this manuscript accessible to a non-TALE specialist.*

I am still convinced that the experimental data provided in this manuscript would have deserved publication in a high impact journal like eLife. Yet, I am afraid the authors have missed their chance to present their data properly.

*I am convinced that professional language editing on its own will not help to improve the manuscript. Rather the authors should team up with other scientist in order to improve the structure of the manuscript.*

*In sum the current manuscript version is only accessible to TALE insiders and thus seems not appropriate for publication in eLife.*

We greatly appreciate your helpful suggestions and comments. I am very grateful to your acknowledging the significance of this work as “would have deserved publication in a high impact journal like *eLife*”. I sincerely apologize for not having revised the manuscript to your satisfaction, mainly because of a likely misunderstanding of *substantial rewriting* in the previous decision letter, which we took as to shorten the manuscript by focusing on data obtained in the rice/*X. oryzae* pathosystem, plus a professional language editing. We have now had the manuscript largely rewritten with the help from my colleague. I hope you would be satisfied with this version. A revised manuscript with tracked changes is submitted as a supplemental data for you to check.

*Reviewer #2:*

*The manuscript "Modifying host basal transcription factor increases resistance to TALE-carrying bacteria in rice" by Yuan et al. is a revised version of paper previously submitted to eLife. The manuscript reports on the recruitment of the general transcription factor TFIIAγ5 to mediate TALE-dependent susceptibility or resistance in rice.*

The manuscript has now been significantly streamlined. Most of the comments made in the previous review have been incorporated and the paper is now easier to read (though still very complex).

We greatly appreciate your helpful suggestions and comments. We have now had the manuscript largely rewritten with the help from my colleagues. I hope you would be satisfied with this version. A revised manuscript with tracked changes is submitted as a supplemental data for you to check.

*Here are few re-iterated (or additional) remarks:*

*It was suggested to simplify qRT-PCR data by only presenting one (or two) of the time points (24 or 48h). This would make data so much easier to read without affecting the message. I do not see the need for a time course and the authors' choice is not clearly justified in their reply.*

We revised Figure 1, Figure 3, Figure 3, Figure 3—figure supplement 1, Figure 3—figure supplement 1, and Figure 4—figure supplement 5 by presenting one time point as suggested by this reviewer.

*Abstract is still "cryptic" and should be improved to better catch reader's attention.*

We extensively revised the Abstract.

Figure 5—figure supplement 1 was modified by deleting two lanes in the Southern blot: Was the Southern redone? If both lanes were merged on the computer, a dashed line should indicate this image manipulation. Or can the authors present raw data for both Southerns? This hold true for any other figure panel.

The two lanes in Figure 5—figure supplement 1 of previous version were merged on the computer and the two lanes were from the lane 1 and lane 4 of the original image. I am sorry for we should have left space between the two lanes. In the revised Figure 5—figure supplement 1, we present the original image that contains four lanes. We checked other images (yeast two-hybrid assays and CoIP assays), and there is no artificial mergence.

*Westerns for Y2H experiments are not shown. This would be important, as the authors tend to use Y2H to measure interaction strengths. Should be done for some key experiments at least.*

Because we do not have specific antibodies for the yeast two-hybrid assays (Y2H), we cannot do Western. However, the results of interaction strength between transcription factor-binding (TFB) region of TALEs and TFIIAγ5 analyzed by Y2H (Figure 2—figure supplement 2) are supported by the results of in planta analyses. To confirm whether a strong interaction cause a higher level of susceptibility than a weak interaction, we swapped the TFB region of TALE pthXo1 with that from other TALEs that showed different interaction strength with TFIIAγ5 in Y2H. These recombinant TALEs were introduced into two *X. oryzae* strains, which do not carry PthXo1. Rice plants showed different levels of susceptibility to the two sets of bacterial strains carrying different recombinant TALE, which is consistent with the Y2H results (Figure 3 and Figure 3—figure supplement 1). Figure 7 presents another set of similar results by using a TALE-free *X. oryzae* strain PH just for review purpose (because our collaborator wishes this TALE-free strain data to first appear in another paper). These in planta data suggest that *X. oryzae*-caused susceptibility requires efficient binding of TALE to TFIIAγ5 by its TFB region to activate rice susceptibility gene expression.

Author response image 1.Effects of the TFB region of TALE pthXo1 on the virulence of *Xoo* strains and on the expression of rice susceptibility gene in rice−*Xoo* interaction.Each bar represents mean (total 30 to 35 leaves from five plants for lesion length; three replicates for gene expression and bacterial growth rate) ± standard deviation. (**A**) Virulence of strain PH and its derivatives carrying pthXol or modified pthXo1 in near-isogenic lines IR24 and IRBB5. PH is an engineered TALE-free strain with the genetic background of strain PXO99^A^, which carries pthXo1. b, significant difference between PH and its derivatives in each rice line at *P* < 0.01. (**B**) Growth of different strains in rice leaves. b, significant difference between 0 day (30 minutes after infection) and 12 days after infection of each strain at *P* < 0.01. (**C**) Expression of susceptibility gene Os*8N3/SWEET11* after infection of different strains. b, significant difference between non-inoculated (ck) and inoculated (at 48 h after infection of a strain) plants in each rice line at *P* < 0.01.**DOI:**
http://dx.doi.org/10.7554/eLife.19605.028

*Reviewer #2 (Additional data files and statistical comments):*

*Statistical analysis of Q-PCR or pathoassays results by t-test is likely incorrect since it only applies to relatively large data sets (ca. >30 biological replicates) with normal distribution. Were both assumptions verified?*

We agree that the numbers of repeats are small for statistical test, but this has been a common practice used in in this sort of studies because of the nature of the experiments. However, I do not quite agree with this reviewer about the requirement of t-test. A t-test is for testing significance of difference between two small samples (Robert Steel and James Torrie 1980 Principles and Procedures of Statistics A Biometrical Approach McGraw-Hill Book Company). If the sample sizes are larger than 30, z-test would be used and the test criterion would approximately follow a normal distribution. I have not seen in the literature any Q-PCR assay being repeated 30 times which is not necessary, neither am I aware any one has assessed the statistical distribution of Q-PCR data. But it seems that there is no reason to doubt that the Q-PCR value would not follow a normal distribution. Nevertheless, the t-test like the case here is useful for having some idea about the likely statistical significance of the differences observed in this kind of experiments, rather than not doing it at all.